# Effect of Al, Ga, and In Doping on the Optical, Structural, and Electric Properties of ZnO Thin Films

**Samuel Porcar** [1] **, Jaime González** [1] **, Diego Fraga** [1,*] **, Teodora Stoyanova Lyubenova** [2] **, Gina Soraca** [3] **and Juan B. Carda** [1]

[1] Escuela Superior de Tecnología y Ciencias Experimentales, Departamento de Química Inorgánica y Organica, Universitat Juame I, 12006 Castellón de la Plana, Spain; saporcar@uji.es (S.P.); jcuadra@uji.es (J.G.); carda@uji.es (J.B.C.)

[2] Joint Research Centre, Directorate C—Energy, Transport and Climate Energy Efficiency and Renewables, ESTI—European Solar Test Installation, European Commission, I-21027 Ispra, Italy; Teodora.LYUBENOVA-STOYANOVA@ec.europa.eu

[3] Escuela de Ciencias Químicas, Universidad Pedagógica y Tecnológica de Colombia (UPTC) de Tunja, Tunja 150003, Colombia; gina.soraca@uptc.edu.co

\* Correspondence: fraga@uji.es

**Abstract:** ZnO thin films with oxygen vacancies and doped with Al, Ga, and In ($Zn_{1-x}M_xO_{1-y}$ (x = 0.03)) have been successfully deposited on soda-lime glass substrates using a simple soft chemical method. The crystalline structure shows a single hexagonal phase of wurtzite with preferred crystal growth along the 002 plane. The surface morphology, characterized by SEM, revealed that the grain shape varies depending on the dopant agent used. Optical measurements displayed an increase in the bandgap values for doped films from 3.29 for ZnO to 3.35, 3.32, and 3.36 for Al, Ga, and In doped films, respectively, and an average transmittance superior to 90% in some cases (in the range between 400 and 800 nm). The electrical response of the films was evaluated with a four-point probe being 229.69, 385.71, and 146.94 $\Omega$/sq for aluminium, gallium, and indium doped films, respectively.

**Keywords:** zinc oxide; thin films; optical properties; electrical properties; photovoltaic applications

## 1. Introduction

Transparent conductive oxides (TCOs) are commonly used in the development of solar cells [1,2], light-emitting diodes, or sensing devices, among others, which normally include ITO ($In_2O_3$:Sn), FTO ($SnO_2$:F), or AZO (ZnO:Al). For their application, thin films need to present low resistance and high transmittance. Among the three mentioned TCOs, ITO displays the best characteristics and is therefore the most widely used [3]. However, it presents some important disadvantages too, for example the high price and the toxicity of the material due to the presence of indium [4]. Despite the fact that FTO presents good characteristics, sometimes it cannot be used due to fluor diffusion between layers [5,6]. However, zinc oxide thin films present good stability, non-toxicity, and are a low-cost material [7].

TCO based doped ZnO films are also widely used as they are relatively cheap [8], can be textured for light trapping, and readily produced for large-scale coatings. They allow tailoring of the absorption in the UV region (transparency in the VIS region) and, when doped with other elements, present good electrical behaviour [9]. Furthermore, ZnO is resistant to oxygen and moisture, has very good optical transparency, and has a flexible host crystal lattice able to accept a variety of dopant substitutions.

Some deposition methods have been reported using different techniques, including the sol-gel method [10], laser deposition [11], magnetron sputtering [12], spray pyrolysis, atomic layer deposition [13,14], or chemical baths [15–17]. The atomic layer deposition is the most commonly used technique for the development of this type of film due to the

high homogeneity and control of the grown film. Despite this, the technique has a low grow rate and requires very controlled atmospheres, which makes the industrial scaling difficult. Methods in which highest quality films are obtained are expensive and difficult to incorporate in less advanced industries; for this reason, it was considered convenient to study the optimization of a spray pyrolysis method using a manual aerograph, due to the simplicity of the process with similar optical and electrical results and the physics processes as an alternative to the ALD process.

The films properties will change when exposed to spray pyrolysis, principally depending on the substrate temperature, atmosphere composition, carrier gas, droplet size, distance between the nozzle, the substrate, cation–anion ratio, and precursor solution concentration [18]. It gives to the technique a wide range of possibilities to improve and change the films' properties.

## 2. Materials and Methods

Doped ZnO was synthesized using a solution-based method. The precursors used were $ZnO(CH_3COO)_2 \cdot 2H_2O$ (99.5% of purity, from PanReac Applichem, Castellar del Valles, Spain), $Al(NO_3)_3$ (98% of purity, from Sigma Aldrich), $Ga(NO_3)_3$ (99.9% of purity, from Sigma Aldrich, Wilson, USA), and $In(NO_3)_3$ (99.9% of purity). The 3% metal-doped ZnO solutions ($Zn_{0.97}M_{0.03}O$) were prepared dissolving $ZnO(CH_3COO)_2 \cdot 2H_2O$ (0.1834 mol) and the dopant precursor $Al(NO_3)_3$, $Ga(NO_3)_3$, or $In(NO_3)_3$ ($5.7 \times 10^{-3}$ mol) in a mixture of ethanol (25 mL) and methanolamine (0.190 mol). Doped and undoped ZnO thin films were deposited by spray pyrolysis system onto soda-lime glass ($1.5 \times 1.5$ cm$^2$). Before the deposition, substrates were cleaned with a 1:2 $HCl:HNO_3$ solution, ultrasonicated in distilled water, and then in ethanol for 5 min each. The distance between the substrate and the spray head nozzle was kept as 20 cm, the flow rate was 1 mL/min and the substrate temperature was raised at 425 °C on a hot plate (surface temperature was measured with a laser thermometer ST80 ProPlus Enchanced), 20 mL of solution was sprayed for each sample. The deposited doped films were then individually annealed under $N_2/H_2$ (5% $H_2$) atmosphere in a tubular furnace. The applied thermal cycle corresponds to a heating velocity rate of 20 °C/min up to a maximum temperature of 400 °C, where it had remained for 30 min followed by a free cooling.

The crystal structure of ZnO doped with Al, Ga, and In, and undoped ZnO thin films (called ZnO:Al, ZnO:Ga, ZnO:In, and i-ZnO, respectively) were studied using an X-ray diffractometer (D4 Endeavor, Burker-ASX) equipped with a Cu K$\alpha$ radiation source. Data were collected by step-scanning from 10° to 80° with a step size of 0.05° 2θ and 3 s counting time per step. The surface morphology was studied using a Scanning Electron Microscopy (SEM) model JEOL 7001F. The layer thickness was determined by the microscopy too, from cross-section micrographs. Optical properties and bandgap energy of the films were conducted by UV–vis–NIR spectroscopy in the wavelength range 200–1000 nm (step size 1 nm) using a Cary 500 Scan Varian spectrophotometer. The transmittance spectra were obtained applying $BaSO_4$ integrating sphere as white reference material. Electrical measurements were carried out by the four-probe method using Ossila T2001A3 to obtain the sheet resistance and resistivity data.

## 3. Results

Well-developed crystalline films of ZnO (intrinsic and doped) were confirmed by the XRD pattern (JCPDS# 79-2 205). XRD patterns of doped ZnO thin films deposited onto soda-lime glass substrate are shown in Figure 1.

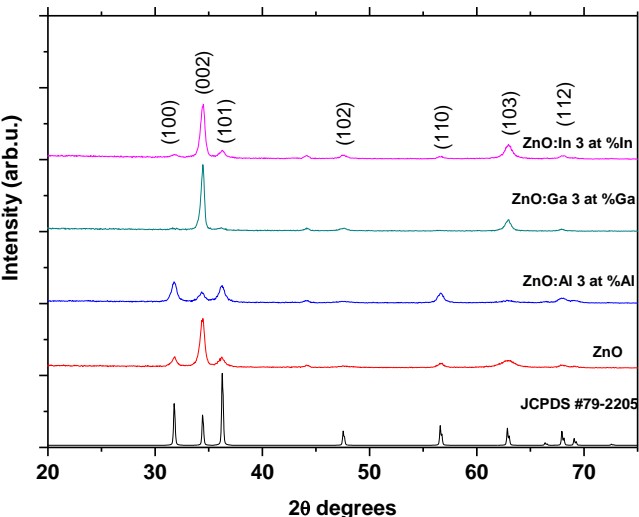

**Figure 1.** XRD spectra of i-ZnO, Al, Ga, and In samples compared with a ZnO pattern.

The crystallite sizes were quantitatively evaluated from XRD data using the Debye–Scherrer equation (Equation (1)) [19]:

$$D = k\lambda / \beta\cos\theta \tag{1}$$

where k is Scherrer constant (0.9 for spherical particles), λ the X-ray wavelength (1.5405 Å), β the peak width of half-maximum, and θ is the Bragg diffraction angle [20]. The resultant particle size for ZnO films is between 12nm and 35.5 nm and becomes shorter for doped films as can be seen in Table 1.

**Table 1.** Optical and electrical properties of doped ZnO grown at soda-lime glass.

| | Voltage (V) | Intensity (mA) | Sheet Resistance (Ω/sq) | Band Gap (eV) | Transmittance(%) | Crystallite Size (nm) | Thickness (nm) |
|---|---|---|---|---|---|---|---|
| i-ZnO | 3 | 0.017 | 794,117.65 | 3.29 | 87.1 | 12–35.5 | 210 |
| $ZnO_1$-x | 0.186 | 0.99 | 829.70 | 3.27 | 92.3 | | |
| $Al^{3+}$ | 0.049 | 0.96 | 229.69 | 3.36 | 91.6 | 14–20 | 180 |
| $Ga^{3+}$ | 0.084 | 0.98 | 385.71 | 3.33 | 86.6 | 14–22 | 200 |
| $In^{3+}$ | 0.032 | 0.98 | 146.94 | 3.35 | 70.3 | 10–25 | 420 |

The surface morphology of the doped ZnO thin films is an important parameter for a good performance in PV devices. Figure 2 shows the surface SEM images of doped ZnO layers deposited onto soda-lime glass, from which we could see apparent particles with different particle sizes and shapes of the four samples.

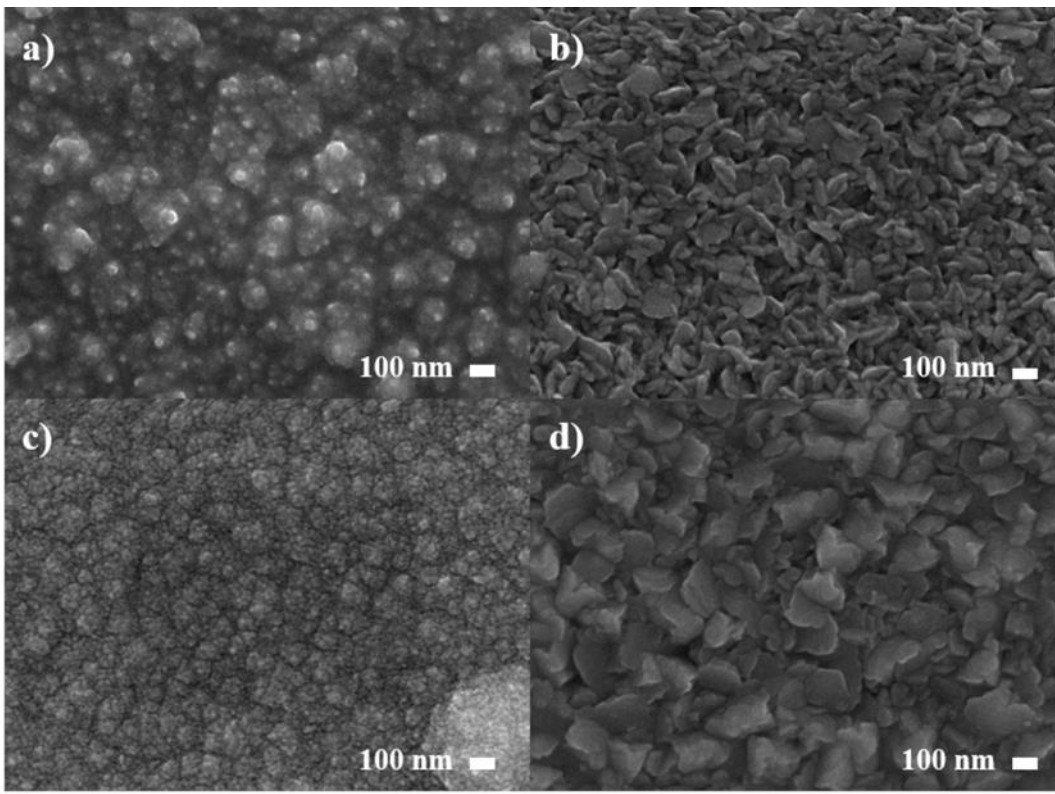

**Figure 2.** Thin films surface morphology (**a**) i-ZnO, (**b**) ZnO:Al, (**c**) ZnO:Ga, and (**d**) ZnO:In.

The samples presented a thickness between 180 nm and 450 nm, as can be observed in Figure 3 where films' cross-section micrographs are shown.

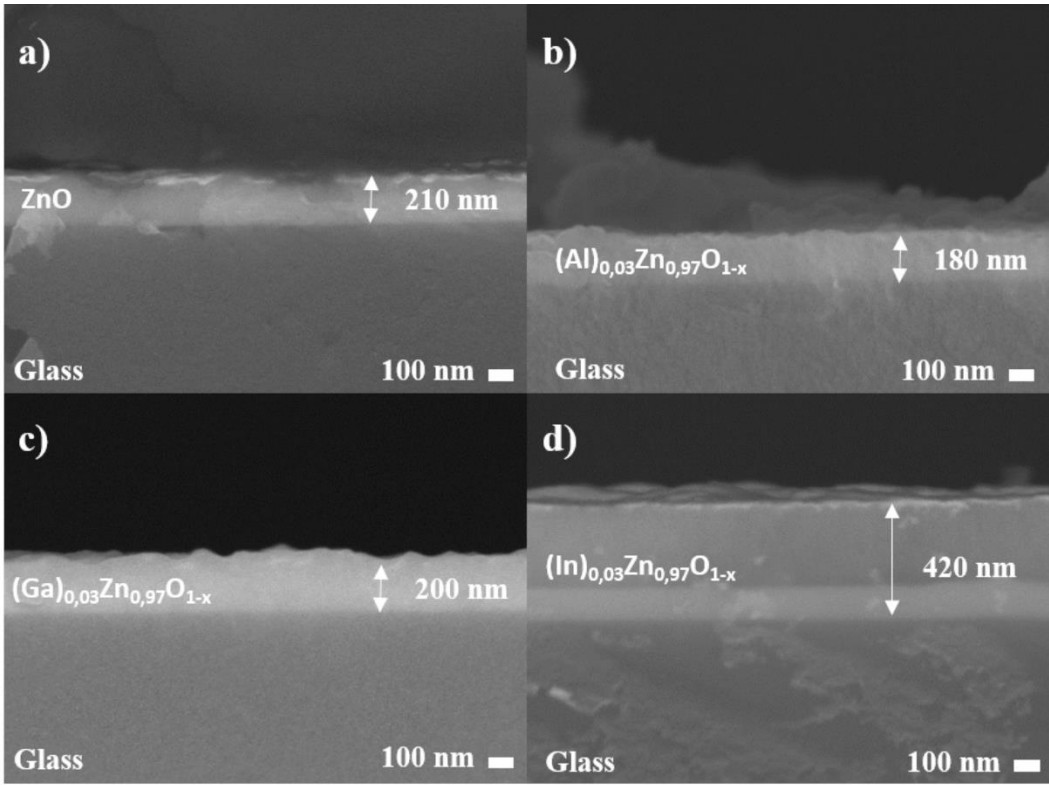

**Figure 3.** Cross section films images (**a**) i-ZnO, (**b**) ZnO:Al, (**c**) ZnO:Ga, and (**d**) ZnO:In.

Figure 4 shows a comparison of the optical transmittance spectra between doped ZnO films and the intrinsic ZnO film deposited on a soda-lime substrate.

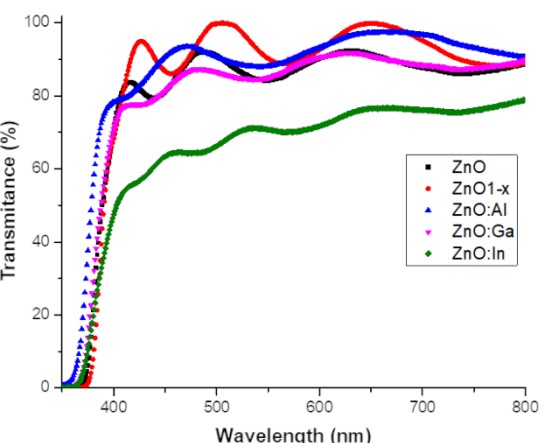

**Figure 4.** Transmittance spectra of ZnO doped and undoped thin films.

Taking these results, bandgap values for each film was also obtained. For this first, the absorption coefficient ($\alpha$) values were calculated using Lambert's Law as the following equation (Equation (2)) [21]:

$$A = 1/t \ln(1/T) \tag{2}$$

where T is the transmittance, and t is the film thickness. An optical bandgap of thin films was estimated using the equation x by the extrapolation of ($\alpha$h$\nu$)2 vs. h$\nu$ (Equation (3)):

$$Ah\nu = A(h\nu - E_g)^{(1/2)} \tag{3}$$

where A is a constant, h$\nu$ is the photon energy, and Eg is the optical bandgap. The optical bandgap of ZnO and ZnO doped thin films was determined by extrapolating the region of the plot to the energy axis where $\alpha2 = 0$, and was 2.

The optical band gap of doped ZnO is shown in Figure 5. The values obtained are 3.29, 3.35, 3.32, and 3.36 eV for i-ZnO, ZnO:Al, ZnO:Ga, and ZnO:In, respectively. The electrical behaviour of samples is shown in Table 1.

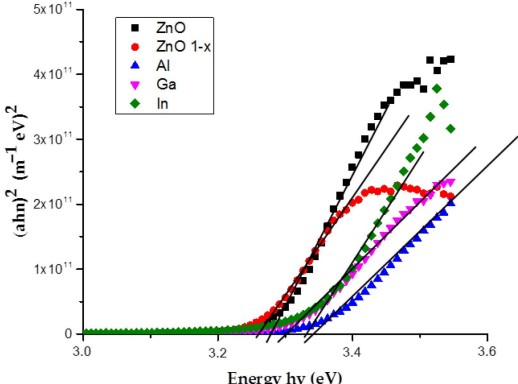

**Figure 5.** ($\alpha$h$\nu$)$^2$ front h$\nu$ curves of i-ZnO and ZnO doped with Al, Ga, and In.

## 4. Discussion

For all the samples, diffraction peaks at 31.8°, 34.5°, 36.2°, 47.6°, 56.6°, 62.9°, 68.0°, and 69.1° have been observed. The peak observed at 34.5° corresponds to the preferred orientation (002) plane of hexagonal structure of ZnO, indicating that all the deposited thin films have the same structure type [22,23]. Any secondary phase is observed, confirming that all

dopants are in the structure. The particle size varies depending on the dopant agent used, for aluminium and indium doped films, more plane particles were observed, however for undoped and gallium doped films the morphology changes appearing spherical particles. In the visible region with wavelengths ranging from 350 nm to 800 nm, samples present an average transmittance in the visible (400 nm to 800 nm) of 91.6%, 86.6%, 70.3%, and 87.1% for Al, Ga, In, and i-ZnO, respectively. All films present a high optical transmittance which decreases with the heaviest doping elements.

Band gap values are larger than the band gap of pure ZnO film (about 3.3 eV) which can be attributed to the Burstein–Moss effect [12], which is caused by the increase in free electron concentration after $M^{3+}$ doping. In addition, a sample of ZnO treated in the tubular furnace ($ZnO_{1-x}$) was studied to see the effect caused by the oxygen vacancies independently; a bandgap decrease was observed due to the oxygen vacancies generated [9]. Furthermore, it can be observed that aluminium and indium doped films, which present a similar morphology, have a higher bandgap than gallium and intrinsic ZnO, which had spherical particles.

As is shown, the sheet resistance decreases when a dopant agent is added. Although generated oxygen vacancies help in the decrease of the sheet resistance too, the presence of the dopants is necessary to have better results, confirming that the good electrical properties were obtained from a combination of the oxygen vacancies generated and the dopant added. The best electrical results come from In and Al doped films, which presented a similar particle morphology.

## 5. Conclusions

Low resistant and highly transparent ZnO doped thin films, used for transparent conductive oxide applications by an easy, low-cost spray pyrolysis technique, were successfully developed. The structural, morphological, optical, and electrical properties of the coating with composition $Zn_{1-x}M_xO_{1-y}$ (x = 0.03) were characterized. All film crystals were grown in a wurtzite structure with preferred c-axis orientation along 002 crystal plane. Layers presented an optical transmittance higher than 90% in the case of aluminium doping. An increase in the energy band gap of doped ZnO was observed; the values go from 3.32 to 3.36 eV, depending on the dopant and morphology, and decrease at 3.27 eV when only oxygen vacancies were generated. Good sheet resistance values of 229.69, 385.71, and 146.94 $\Omega$/sq were obtained for ZnO:Al, ZnO:Ga, and ZnO:In thin films.

**Author Contributions:** Conceptualization, J.B.C. and D.F.; methodology, S.P.; software, T.S.L.; validation, T.S.L.; investigation, S.P., G.S., and J.G.; writing—original draft preparation, S.P.; writing—review and editing, D.F. and T.S.L.; supervision, J.B.C. and D.F.; project administration, J.B.C.; funding acquisition, J.B.C. All authors have read and agreed to the published version of the manuscript.

**Funding:** This research was funded by Spanish Ministry of Economy and Competitiveness under the program Programa Estatal de I+D+I orientada a los retos de la sociedad (IGNITE Project Ref. ENE2017-87671-C3-3-R) and program Proyectos de I+D+i» de los Programas Estatales de Generación de Conocimiento y Fortalecimiento Científico y Tecnológico del Sistema de I+D+i y de I+D+i Orientada a los Retos de la Sociedad, en el marco del Plan Estatal de Investigación Científica y Técnica y de Innovación 2017-2020 (Ref. PID2020-116719RB-C43).

**Institutional Review Board Statement:** Not applicable.

**Informed Consent Statement:** Not applicable.

**Acknowledgments:** We also appreciate the characterization assistance of the Central Service of Scientific Instrumentation (SCIC) at the University Jaume I. We also thank the group of F. Fabregat (INAM-UJI) for the analysis and characterization assistance.

**Conflicts of Interest:** The authors declare no conflict of interest. The funders had no role in the design of the study; in the collection, analyses, or interpretation of data; in the writing of the manuscript, or in the decision to publish the results.

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
