# Peer review of "Effect of Al, Ga, and In Doping on the Optical, Structural, and Electric Properties of ZnO Thin Films"

_applsci, doi:10.3390/app112110122_

Round 1

Reviewer 1 Report

Porcar et al. investigated optical, structural and electrical properties of ZnO thin films using Al, Ga an In dopants. I recommend manuscript to accept after author should clarify these issues.

  • Author should add recent studies related to this work in the introduction section.
  • The results of optical measurements are different in abstract and discussion sections.
  • In abstract, Aluminium, Galium and Indium should be changed with aluminum, gallium, and indium.
  • In materials and methods, please follow units in uniform in the manuscript just like 25 ml and 1mL/min.
  • In discussion, page 6, replace oxigen with oxygen

Author Response

  • Point 1: Author should add recent studies related to this work in the introduction section.
  • Response 1: We added information about AZO recent studies in the introduction section.
  • Point 2: The results of optical measurements are different in abstract and discussion sections.
  • Response 2: We solved it
  • Point 3: In abstract, Aluminium, Galium and Indium should be changed with aluminum, gallium, and indium.
  • Response 3: We changed it
  • Point 4: In materials and methods, please follow units in uniform in the manuscript just like 25 ml and 1mL/min.
  • Response 4: It was done
  • Point 5: In discussion, page 6, replace oxigen with oxygen.
  • Response 5: It was modifed

Reviewer 2 Report

In the paper of Samuel Porcar et al. authors described the influence of doping atoms on properties of zinc oxide layers. The comparison character of the paper is very important prom the application point of view. However, the current version of papers need major revision before its publication:

  • Introduction - authors should enhance the introduction part of the paper about short description of ZnO deposited by atomic layer deposition method commonly used in microelectronics
  • line 55 - I do not understand the information provided by authors: (5.7·10-3 mol)
  • 2. Materials and Methods  - there is no information about the time of ZnO deposition process. Was the time the same for different samples?
  • 2. Materials and Methods - did the samples annealed together in the tube furnace or one by one?
  • lines 70-71 - Thickness determination using only SEM imaging is burdened with some error. Did you consider spectroscopic ellipsometry for precise determination of ZnO layer thickness?
  • Electrical characterization of obtained ZnO layers is rather poor. Authors provided only sheet resistance results. What about carrier concentration and their mobility (Hall measurements). Did you consider application of ECV measurement which provide depth dependent carrier concentration?
  • Figure 1 caption - what means DRX? Did authors mean XRD?
  • lines 85 vs 87: Scherre vs Sherrer
  • Please include Table 1 just after first mention
  • Surface morphology was investigated using SEM, however authors do not provide any quantitative parameter describing the morphology. Are you able for example determine RMS from AFM measurement?
  • What is a reason of different thickness of investigated samples?
  • lines 115-117 - the description of band gap determination from Tauc plot is confusing
  • Why authors do not used photoluminescence for determination of band gap?

Author Response

Point 1: Introduction - authors should enhance the introduction part of the paper about short description of ZnO deposited by atomic layer deposition method commonly used in microelectronics.  

Response 1: We added extra information about ALD and spray pyrolisis tecnhiques.

Point 2: Line 55 - I do not understand the information provided by authors: (5.7·10-3 mol)

Response 2: It was modified

Point 3: Materials and Methods  - there is no information about the time of ZnO deposition process. Was the time the same for different samples?

Response 3: We had the information about the flow rate (ml/min) and we added the quantity of precursor solution sprayed (ml) in order to know the time and the amount of solution used.

Point 4: Materials and Methods - did the samples annealed together in the tube furnace or one by one?

Response 4: We clarified it in the experimental section(The deposited doped films were then individually annealed...).

Point 5: lines 70-71 - Thickness determination using only SEM imaging is burdened with some error. Did you consider spectroscopic ellipsometry for precise determination of ZnO layer thickness?

Response 5: We agree with your consideration but nowadays we can't do this type of determination.

Point 6: Electrical characterization of obtained ZnO layers is rather poor. Authors provided only sheet resistance results. What about carrier concentration and their mobility (Hall measurements). Did you consider application of ECV measurement which provide depth dependent carrier concentration?

Response 6: We agree with you. We are considering this possibility. For future studies we will try to application of ECV mesurements and Hall measurements.

Point 7: Figure 1 caption - what means DRX? Did authors mean XRD?

Response 7: It has been changed.

Point 8: lines 85 vs 87: Scherre vs Sherrer

Response 8: We corrected it

Point 9: Please include Table 1 just after first mention

Response 9: Done

Point 10: Surface morphology was investigated using SEM, however authors do not provide any quantitative parameter describing the morphology. Are you able for example determine RMS from AFM measurement?

Response 10: We are not able to determine RMS from AFM measutement.

Point 11: What is a reason of different thickness of investigated samples?

Response 11: As we mentioned in the introduction section we used a handmade sprayed method so is difficult control exactly the thickness and sometimes there are differences itn the thickness. Also, there are another external factors such as humidity or air flow rate that can influence in the thickness.

Point 12lines 115-117 - the description of band gap determination from Tauc plot is confusing

Response 12: In table 1 there are the bang gap values for each sample.  The description of band gap determinations from Tauc plot was made according to the bibliography.

Point 13. Why authors do not used photoluminescence for determination of band gap?

Response 13. Phototlouminescence is a great option to determine de bang pap, however we did the band gap measurment with UV-Vis curves and Tauc plot. We think that both tenchniques are able to determine the band gap.

Round 2

Reviewer 2 Report

I apprecite authors correction. My decision is accept the paper.